# A Transcriptomic Approach to Elucidate the Mechanisms of Gefitinib-Induced Toxicity in Healthy Human Intestinal Organoids

**DOI:** 10.3390/ijms23042213

**Published:** 2022-02-17

**Authors:** Daniela Rodrigues, Bram Herpers, Sofia Ferreira, Heeseung Jo, Ciarán Fisher, Luke Coyle, Seung-Wook Chung, Jos C. S. Kleinjans, Danyel G. J. Jennen, Theo M. de Kok

**Affiliations:** 1Department of Toxicogenomics, GROW School for Oncology and Developmental Biology, Maastricht University, 6229 ER Maastricht, The Netherlands; j.kleinjans@maastrichtuniversity.nl (J.C.S.K.); danyel.jennen@maastrichtuniversity.nl (D.G.J.J.); t.dekok@maastrichtuniversity.nl (T.M.d.K.); 2Crown Bioscience Netherlands B.V., J.H. Oortweg 21, 2333 CH Leiden, The Netherlands; bram.herpers@crownbio.com; 3Simcyp Division, Certara UK Limited, Sheffield S1 2BJ, UK; sofia.ferreira1@astrazeneca.com (S.F.); emily.jo@certara.com (H.J.); ciaran.p.fisher@gsk.com (C.F.); 4Boehringer Ingelheim International GmbH, Pharmaceuticals Inc., Ridgefield, CT 06877, USA; luke.coyle@boehringer-ingelheim.com (L.C.); seung.chung@boehringer-ingelheim.com (S.-W.C.)

**Keywords:** gefitinib, human intestinal organoids, molecular mechanisms, transcriptomics, toxicity

## Abstract

Gefitinib is a tyrosine kinase inhibitor (TKI) that selectively inhibits the epidermal growth factor receptor (EGFR), hampering cell growth and proliferation. Due to its action, gefitinib has been used in the treatment of cancers that present abnormally increased expression of EGFR. However, side effects from gefitinib therapy may occur, among which diarrhoea is most common, that can lead to interruption of the planned therapy in the more severe cases. The mechanisms underlying intestinal toxicity induced by gefitinib are not well understood. Therefore, this study aims at providing insight into these mechanisms based on transcriptomic responses induced in vitro. A 3D culture of healthy human colon and small intestine (SI) organoids was exposed to 0.1, 1, 10 and 30 µM of gefitinib, for a maximum of three days. These drug concentrations were selected using physiologically-based pharmacokinetic simulation considering patient dosing regimens. Samples were used for the analysis of viability and caspase 3/7 activation, image-based analysis of structural changes, as well as RNA isolation and sequencing via high-throughput techniques. Differential gene expression analysis showed that gefitinib perturbed signal transduction pathways, apoptosis, cell cycle, FOXO-mediated transcription, p53 signalling pathway, and metabolic pathways. Remarkably, opposite expression patterns of genes associated with metabolism of lipids and cholesterol biosynthesis were observed in colon versus SI organoids in response to gefitinib. These differences in the organoids’ responses could be linked to increased activated protein kinase (AMPK) activity in colon, which can influence the sensitivity of the colon to the drug. Therefore, this study sheds light on how gefitinib induces toxicity in intestinal organoids and provides an avenue towards the development of a potential tool for drug screening and development.

## 1. Introduction

Gefitinib is a chemotherapeutic agent that belongs to the group of tyrosine kinase inhibitors (TKIs) and that selectively inhibits the activity of receptor tyrosine kinases (RTKs) [1]. In the case of gefitinib, it selectively inhibits epidermal growth factor receptor (EGFR) kinase [2] via competitive inhibition of ATP at the enzyme’s catalytic binding site [3]. Gefitinib is especially used in the treatment of locally advanced or metastatic non-small cell lung carcinoma (NSCLC), particularly in cases that derive from mutations in the EGFR gene [2,4]. This drug has also shown efficacy in the treatment of cutaneous squamous cell carcinoma and advanced cervical cancer [5,6]. After administration, gefitinib is extensively metabolized in an oxidation process that facilitates excretion of the compound and in which the enzymes CYP3A4/5 play an important role [4].

Gefitinib is generally well-tolerated despite some reported adverse effects that can affect approximately 30% of patients. The known side effects most commonly include skin rashes and diarrhoea (30%), and less commonly nausea, vomiting and stomatitis/mucositis (10–25%) [2,7]. Due to the severity of the gastrointestinal (GI) symptoms among patients taking gefitinib, these patients are required to reduce or even stop their treatment, which hampers their chance of survival. Despite the promise and success of gefitinib in the treatment of cancer as compared with other chemotherapeutics, the mechanism underlying gefitinib-driven intestinal toxicity is not yet fully understood. Studies on the intestinal toxicity induced by gefitinib are still very limited as they focus only on efficacy, side effects and intestinal growth inhibition [8,9,10,11,12,13] without investigating the molecular responses underlying those adverse events. No previous studies have examined the gene expression responses of intestinal cells towards gefitinib. For these reasons, this study aimed at generating new transcriptomic data to improve understanding about the molecular mechanisms involved in gefitinib-induced intestinal toxicity. 

In this study, three-dimensional (3D) innovative cell models of human colon and small intestine (SI) were established as described in our previous work [14] due to their potential in drug-response modulation. The establishment of 3D organoid systems has exponentially increased in recent years as they replicate in vivo cellular organization, behaviour, and cell-environment interactions more closely than conventional 2D cultures [15,16]. 3D intestinal organoids are no exception as they have shown to possess key features of human in vivo intestinal tissue, namely crypt-like structures [17,18]. These features have improved the understanding of tissue/organ development, homeostasis, and diseases [19,20,21], particularly in cancer modulation and anti-cancer drug research. Several studies on cancer research have applied 3D organoids to investigate tumorigenesis, cancer progression and therapeutic screening including brain [22], lung [23], breast [24] or GI organoids [25]. Therefore, 3D cell culture technologies are promising alternatives to 2D systems in pre-clinical drug development and drug-specific responses studies [15]. 

Here, we applied 3D colon and SI organoids to investigate gefitinib transcriptomic responses and gain new insights on the molecular mechanisms of the drug toxicity. Responses in colon and SI tissues were compared to establish if their different cell composition, dynamics and function is reflected by distinct gene expression profiles. The exposure concentrations for the in vitro experiments with gefitinib were based on physiologically based pharmacokinetic (PBPK) modelling and simulation to guarantee that the in vitro concentrations of gefitinib reflect patients’ dose regimens during chemotherapy [14,26]. Intestinal organoids were exposed to four different concentrations (0.1, 1, 10 and 30 µM) for three days. After exposure, samples were evaluated for cytotoxicity and characterized for structural/morphological changes by image analysis, which can be associated with gefitinib’s mode of action and transcriptomic data. In-depth quantitative RNA-sequencing was performed to investigate changes in the gene expression profiles of treated organoids. We aim to identify differentially expressed genes (DEGs) and the most relevant affected biological pathways that, when compared with functional endpoints, can provide new molecular markers associated with gefitinib effects on the intestinal tissue. These results could be of future relevance in the development and screening of new TKIs and to better predict potential intestinal damages. In the context of the Translational Quantitative System Toxicology (transQST) project [26], the new transcriptomic data will be applied in a QST model to predict GI toxicity caused by drugs. This is a fundamental contribution to transQST as the project aims at improving the understanding of drug safety by developing tools that facilitate the assessment of safety profiles of potential medicines before clinical testing [26]. 

## 2. Results

### 2.1. VIVD and PBPK-Based Gefitinib Exposure In Vitro Nominal Concentrations

The predictive performance of the gefitinib human PBPK model was validated using observed total plasma concentrations from peer-reviewed literature [27], shown in Figure 1. Observed total plasma concentrations of gefitinib were generally captured within the 95% confidence interval of the simulated plasma concentration-time profile. The validated gefitinib model was then used to predict gefitinib PK after the therapeutic regimen of one daily 250 mg oral dose. The corresponding PK profile of free gefitinib in plasma and jejunum enterocytes are shown in Figure 2 and steady state PK are detailed in Table 1. Free gefitinib concentrations in vivo were of interest as free compound drives the response. Steady state plasma concentrations are achieved after 10 days of one daily dosing of 250 mg oral gefitinib [28]. In plasma, the maximum free concentration reached at steady state (C_max,ss_) was predicted as 0.03 µM and average concentration at steady state (C_av,ss_) was predicted as 0.02 µM. In enterocytes, the predicted free C_max,ss_ and C_av,ss_ was 15.38 µM and 7.69 µM, respectively. Fraction of unbound gefitinib in enterocytes was assumed to be 1, i.e., gefitinib is completely unbound in enterocytes. This assumption is likely an overestimation knowing that gefitinib is highly protein bound in plasma (f_u_ = 0.064). However, in the absence of tissue specific data, this assumption allows the in vitro nominal concentrations selected based on predicted unbound in vivo enterocyte concentrations to reach the highest possible therapeutic exposure.

Intracellular gefitinib concentrations in vitro were predicted using VIVD that simulated the human intestinal organoid assay conditions (Appendix A). The results of predicted intracellular gefitinib concentrations in a range of nominal concentrations at steady state are presented in Table 1. The predicted intracellular concentration to nominal concentration ratio was 91.37, which is in line with gefitinib as a highly lipophilic compound (LogP_ow_ = 4.15) [29]. A nominal concentration of 0.1 µM was predicted to correspond with an intracellular concentration of 9.14 µM, which is comparable to the predicted in vivo jejunum enterocyte C_av,ss_ of 7.69 µM from therapeutic dosing. Therefore, by combining PBPK modelling and VIVD, an in vitro testing strategy that achieved therapeutic tissue exposure was informed. 

### 2.2. Evaluation of Viability and Apoptosis 

Cell viability was established for colon and SI organoids exposed to 0.1 µM–30 µM gefitinib based on measurements of ATP levels and apoptosis reflected by caspases 3/7 activation (Figure 3). In colon organoids, there was no significant effect observed in either cell viability or caspase activation for the lower concentration (0.1 µM) as compared to the controls (Figure 3A,B). A decrease in viability was first noticeable at 1 µM, and continued to decrease significantly at 10 µM and 30 µM, in a concentration and time-dependent manner. At the highest concentration and longest time point, viability decreased by more than 60% (Figure 3A). On the other hand, caspase activation only presented a significant increase after 10 µM exposure, particularly after 48 h and 72 h (Figure 3B). The experimental condition of 30 µM gefitinib at 72 h indicated a significant activation of apoptotic processes in the colon organoids (600% caspase activation). 

In the case of SI organoids (Figure 3C,D), no significant effect was found in either assay for the lower dose, except at 72 h. At 1 µM, viability starts to decrease with concentration and time. Similar to the effects in colon organoids, the greatest decrease in viability was observed at the highest concentration and longer time points, declining by approximately 80% (Figure 3C) In turn, there was no significant change in caspase activation until 30 µM exposure, particularly at 48 h and 72 h, which showed the largest increase in caspase activation (about 400%) relative to the controls (Figure 3D).

### 2.3. Evaluation of Morphological Changes by Image-Based Analysis

Effects of gefitinib on the morphology and structure of the human organoids was evaluated, namely organoid size (growth inhibition), roundness (loss of bud-like differentiated structure) and percentage of cell death (cell cycle arrest and apoptosis activation) (Figure 4), nuclei size and number (growth/cell cycle inhibition). Size of the colon organoids (Figure 4A) did not change significantly upon exposure to gefitinib, whereas the size of SI organoids (Figure 4B) was significantly reduced in all treatment conditions, especially after 48 h and 72 h exposure. Moreover, SI organoids appeared more sensitive to changes in roundness with increases observed at 48 h with the lowest drug concentration, whereas colon organoid roundness only began to increase after exposure to 1 µM and showed an overall lower degree of change (Figure 4C,D). As with the percentage of cell death, roundness was more significantly increased after exposure to 10 µM for 72 h in both organoids (Figure 4E,F). However, this increase was more significant in colon than in SI organoids, particularly after exposure to the highest concentration at all time points, which is in line with the caspase 3/7 activation described above. 

In addition, exposure of colon organoids to gefitinib induced a modest time and concentration-dependent decrease of the nucleus size, with significant decreases in the number of nuclei per organoid at 48 h and 72 h (Appendix A), which corresponds to an increase in cell death. In SI organoids, exposure to gefitinib led to a modest reduction of both nucleus size and number per organoid, being more significantly decreased at 30 µM and 72 h (Appendix A), indicating an induction of cell death at later time points. Furthermore, Appendix A show microscope images of colon and SI organoids, comparing the morphological changes between the controls and gefitinib treatments.

In summary, the morphological changes reflected the results obtained for the viability and caspase 3/7 assays, as gefitinib inhibited cell growth in both organoids and led to cell death mainly in colon organoids in a time- and concentration-dependent manner. Size and roundness of SI organoids were in line with the decrease in viability, whereas in colon organoids only the increased roundness was in line with viability assay. This could suggest that colon organoids maintained their size throughout the exposure without growing further, rather than becoming smaller. The percentage of dead cells was also in line with the caspase activation assay, being more increased in colon than in SI organoids, thus suggesting activation of apoptotic events. 

### 2.4. mRNA Sequencing Data Processing and Pathway Analysis

After the exposure of human colon and SI organoids to gefitinib, mRNA was isolated and sequenced. The obtained sequences were aligned with the human genome and ranged between 72.3% and 36.4% in colon samples and between 81.2% and 38.7% in SI samples. For normalization purposes, samples that yielded a number of read counts below 5 million were eliminated. As a result, 2 colon samples, from different treatment conditions, were not included in the analysis. All samples derived from SI organoids presented more than 5 million read counts. After applying the Bonferroni correction [30] and considering adjusted *p* value <0.05, the number of DEGs tends to increase with gefitinib concentration and time of exposure. Furthermore, the number of DEGs was significantly higher in the colon organoids as compared to the SI in all treatment conditions, except for the lowest concentration (0.1 µM).

After data processing, PCA score scatter plots were generated to observe how the treatment conditions with gefitinib would affect the samples derived from colon and SI and to further explore differences between gene expression of treated and untreated organoids (Figure 5). Looking at the colon organoids PCA plot (Figure 5A), Principal component 1 (PC1) was more correlated with the effect of concentration, and PC2, with the effect of time. It was observed that the controls were clustering together along with the lower concentration of gefitinib. There was also a clear separation between the different concentrations of gefitinib, although the two higher concentrations also clustered together on the right. Furthermore, this separation increased across time, as samples at 24 h and 48 h were closer and at 72 h there was a clearer separation. In the PCA plot obtained for SI organoids (Figure 5B), PC1 was also more correlated with the effect of concentration, and PC2 with the effect of time from 24 h to 48 h and 72 h. There was a clear separation between 24 h and later time points. Moreover, 0.1 µM was separated from the rest of the concentrations, particularly at 48 h and 72 h, whereas higher concentrations appeared in a same cluster on the right and separation between them was more evident between 24 h and 48 h/72 h. Therefore, when comparing the distribution of samples from colon to SI organoids, in colon there was a clearer separation of samples in a concentration (PC1) and time (PC2) dependent manner. Similarly, in SI, the effect of concentration could be observed in PC1, whereas in PC2, there seemed to be an influence of time. 

### 2.5. Pathways and DEGs Affected by Gefitinib

The effect of gefitinib on gene expression was observed in both organoids in a concentration- and time-dependent manner, particularly after exposure to 1µM. The lowest concentration (0.1 µM) did not significantly disturb the biological pathways and gene expression, in line with the results obtained in the cytotoxicity assays. The affected pathways and related DEGs were either on-target toxicity mechanisms (Table 2), i.e., modulated by gefitinib in an EGFR signalling inhibition dependent manner, or off-target (Appendix A), i.e., not directly related to EGFR signalling inhibition but by other toxicity mechanisms triggered by gefitinib and its metabolites. 

#### 2.5.1. Signalling Transduction Pathways

The biological processes that stood out as significantly affected were signalling transduction pathways triggered by the EGFR, including EGF/EGFR, PI3K-Akt and MAPK signalling cascades, signalling by MET and signalling by NOTCH (Table 2), which are in line with the known mechanism of action of the drug (i.e., EGFR inhibition) [2]. In the colon these pathways were more affected than in SI since overall q-values were lower and the number of DEGs involved were higher for colon organoids. In view of the fact that the activity of these signalling pathways is dependent on phosphorylations and that mRNA sequencing data does not provide information on protein level changes, we next looked into the DEGs whose expression depends on those signalling cascades and pathways in which the DEGs are involved, using Reactome, Kyoto Encyclopaedia of Genes and Genomes (KEGG) and WikiPathways as references (Table 3).

Starting with the DEGs regulated by the EGFR pathway, the majority of genes found to be significantly affected are involved in cell cycle progression (phase G1/S) and proliferation, namely *AURKA* (also important in DNA repair), *CCND1* (cyclin D1), *c-MYC*, *c-FOS* and *MYBL2*, with the only exception of *BCL2L11*, which is involved in regulation of apoptosis, being mostly pro-apoptotic. Additional genes *PEPCK* (glucose metabolism) and *CCNG2* (cell cycle arrest), were also found to be significantly affected. These two genes are regulated by FOXO-mediated signalling, which is closely related to EGFR activity. Expression levels of *AURKA* were decreased in colon at 24 h, whereas in SI, only after 72 h exposure to 30 µM. *CCND1* and *c-MYC* presented similar profile changes, being more downregulated at 24 h in colon and after 48 h exposure in SI. Expression levels of *c-FOS* and *PEPCK* were only found significantly changed in the colon, the first being decreased in all treatment conditions and more at 24 h, and the second being decreased only after 48 h and 72 h exposure to 10 and 30 µM. On the other hand, *MYBL2* and *CCNG2* were only observed in SI. Expression levels of *MYBL2* tended to decrease across treatment conditions with the largest decrease at 24 h, 30 µM, while expression levels of *CCNG2* tended to increase, being more upregulated at 72 h. In turn, pro-apoptotic gene *BCL2L11* was found upregulated in both organoids. Expression levels of this gene were more increased at 24 h in colon, whereas in SI they did not change much throughout the exposure. 

Furthermore, PI3K signalling related gene *AREG* (cell growth) was found downregulated in both organoids, but more in the colon. Expression levels of *FGF19*, which is involved in several cellular processes (growth, tissue repair, and metabolism) tended to decrease across concentration and time in colon, and in SI, they were decreased only at 1 µM, all time points, and 10 µM, 24 h. Another interesting gene, *MUC20* (suppressor of signalling by MET), was found upregulated in both organoids but in different treatment conditions. In colon, *MUC20* was only upregulated after 72 h exposure to 10 and 30 µM, and in SI after exposure to 1 µM for 24 h and 48 h. In colon organoids, two other genes associated with signalling pathways were significantly upregulated, namely *TLR2* (inflammation signals) and *DTX3* (signalling by NOTCH) whereas, in SI organoids, they were not significantly affected by the drug.

#### 2.5.2. p53 Signalling Pathway

Additionally, p53 signalling pathway, which regulates gene expression during stress conditions [36], was also significantly affected, presenting lower q-values for SI organoids but only significant after 72 h exposure to 1 µM, whereas in the colon q-values decreased after 24 h exposure to 1 µM (Table 1). More DEGs were significantly affected by the p53 signalling pathway in the colon upon exposure to gefitinib, all of which being involved in either gefitinib on-target processes, namely cell cycle arrest, cell growth, DNA repair, and apoptosis (Table 3), and off-target processes such as metabolism (Appendix A). Starting with genes in common between colon and SI organoids, cell cycle arrest genes *BTG2* (G1/S phase) and *SFN* (G2/M phase) were found upregulated and downregulated, respectively, for all treatment conditions, but more significantly in colon. Apoptotic gene *PERP* was significantly downregulated after 72 h exposure, especially in colon for the higher concentrations. The gene *PRKAB1*, an encoding gene of the regulatory subunit of the activated protein kinase (AMPK), was upregulated in colon after 72 h exposure whereas in SI, it was downregulated. Additionally, in colon organoids, the *PCNA* (DNA repair), *SUSD6* and *BAX* (apoptosis), and *SESN1* (response to DNA damage and oxidative stress) genes were significantly upregulated, at different treatment conditions. In turn, the *CCNG1* (cell cycle), *TNFRSF10* (apoptosis) and *FUCA1* (metabolism of glycoproteins and glycolipids) genes were significantly downregulated after 72 h exposure to 30 µM. Only one gene was significantly changed in SI but not in colon, namely *DDB2* (DNA repair), being only upregulated after 48 h exposure to 30 µM. 

#### 2.5.3. Cell Cycle and Cellular Senescence

In addition to signalling transduction pathways, other important gefitinib on-target cellular processes were investigated to further understand gefitinib effects on the organoids. Firstly, cell cycle was not significantly affected overall, as well as DNA synthesis/replication, except for regulation of mitotic cell cycle at 24 h (Table 2). This was more evident in colon (lower q-values) as also more genes had their levels decreased upon exposure. In contrast, mechanisms of cell cycle arrest prevailed with *CCND1* and *c-MYC* downregulated as described above but expression levels of cell cycle inhibitors p27 and p21 were not found significantly affected in both organoids. However, cellular senescence, which consists of an irreversible cell cycle arrest, was affected in both organoids, but particularly earlier in colon than in SI. Induction of apoptosis was not significantly modulated by the exposure until later treatment conditions in both organoids, in which q-values were lower in SI (Table 2). Nevertheless, expression levels of pro-apoptotic genes *BCL2L11*, *BAX*, *TNFSF10*, *PERP* and *SUSD6* were more significantly increased in colon. Likewise, anti-apoptotic genes *BCL2L1*, *MCL1* and *BIRC3* were more significantly downregulated in colon.

#### 2.5.4. Cell Motility and Adhesion 

We next looked into expression of genes associated with cell motility and adhesion since levels of cell adhesion molecules, namely E-cadherin, β1-integrin and ZO-1, were reported as significantly inhibited in previous studies [33,34]. In our study, the genes encoding for E-cadherin (*CDH1*) and β1-integrin (*ITGB1*) were significantly downregulated, the first in both organoids and the second only in colon organoids (Table 3). Levels of ZO-1 encoding gene *TJP1* were not significantly affected. In addition, genes *TNS3* and *TNS4*, involved in signal transduction pathways for cell motility/migration and whose expression is regulated by EGFR activity, were found upregulated and downregulated, respectively. Under normal circumstances, EFG activates *TNS4* and inhibits *TNS3*, so exposure to gefitinib reversed their expression profile. 

#### 2.5.5. Immune Responses

The increase in the expression of the pro-inflammatory cytokines interleukin (IL)-6 and IL-25 has been reported [33]. In our study, IL-6, IL-1 and IFN-ɣ signalling pathways were also significantly affected in colon organoids at 72 h, 10 and 30 µM based on q-values. Despite that, expression levels of cytokines were not significantly affected except for IL-1, which was found downregulated. In SI organoids, immune responses and expression levels of cytokines were not found significantly changed. 

In summary, the results show that gefitinib on-target toxicity signalling transduction pathways, namely EGFR, PI3K, MET and NOTCH signalling, were significantly affected, which consequently affected the expression of genes directly regulated by these signalling cascades including *AURKA*, *CCND1*, *c-MYC*, *c-FOS* and *FGF19* (Table 3). Most of these genes were downregulated over time and concentrations of gefitinib, except *c-FOS* in the SI organoids. Regarding expression of p53 gene, it was not significantly changed but DEGs influenced by p53 signalling pathways, as well as on-target toxicity mechanisms, were affected across treatment conditions, namely cell cycle arrest-related genes *BTG2* and *CCNG1* (upregulated), and *CCNG1* and *SFN* (downregulated), apoptotic genes *BAX*, *BCL2L11* and *SUSD6* (upregulated), *PERP* and *TNFRSF10* (downregulated), and impairment of cell growth and DNA repair-related genes *DDB2*, *PCNA* and *SESN1* (upregulated). Gefitinib on-target DEGs involved in cell adhesion and motility (E-cadherin, β1-integrin and TNS4) were significantly downregulated across treatment conditions, which reflects additional damaging effects caused by gefitinib. Therefore, expression levels of the genes listed above (Table 3) and represented in Figure 6 were in line with the anti-proliferative effects of the drug, cell cycle arrest, DNA repair, and cell motility/migration. Immune signalling pathways and levels of pro-inflammatory cytokines, an off-target toxicity mechanism, were only significantly modulated in colon. Furthermore, results also indicate that even though similar mechanisms are perturbed by gefitinib in colon and SI cells, genes and their expression levels are distinct. 

#### 2.5.6. Metabolic Pathways and Cholesterol Biosynthesis

Metabolism is considered as an off-target toxicity mechanism induced by gefitinib and was significantly affected in both organoids. The effect was stronger in colon than in SI (Appendix A) and at different treatment conditions. One exception was drug metabolism by cytochrome P450, which was more relevant for SI organoids throughout all treatment conditions, with q-values lower than those observed for colon. This could indicate a novel finding on a tissue-specific response to the drug by SI organoids. Indeed, looking at the expression of the CYP450 genes involved in gefitinib metabolism, *CYP3A4* was only found significantly downregulated in colon (*p*-value = 5.1 × 10^−16^) whereas *CYP3A5* was only found significantly upregulated in SI (*p*-value = 3.9 × 10^−30^). 

Metabolism of proteins and amino acids were similarly affected, with q-values decreasing across concentrations, at earlier treatment conditions in SI (48) than in colon organoids (72 h). Key metabolic pathways for energy production, including glycolysis, TCA cycle, pyruvate metabolism, respiratory electron chain and ATP synthesis, and metabolism of lipids, were observed and q-values were overall lower for colon organoids and more significant at earlier treatment conditions than for SI (Appendix A). TCA cycle and ATP synthesis were not significantly affected in SI organoids. Expression levels of DEGs involved in these energy generation pathways were decreased in both organoids, except for DEGs involved in metabolism of lipids. Interestingly, lipogenesis related genes were downregulated in colon but upregulated in SI organoids. Likewise, cholesterol biosynthesis was differently affected in colon and SI. While this pathway was affected in colon organoids after 1 µM exposure and q-values tended to increase across concentrations, in SI q-values were only significantly increased after exposure to 30 µM. DEGs involved in cholesterol biosynthesis also presented opposite direction of expression, thus downregulated in colon and upregulated in SI organoids (Appendix A).

Next, we looked into possible differences related to the endoplasmic reticulum (ER) stress responses and key regulatory genes of metabolic balance that could aid in understanding the distinct expression of DEGs involved in cholesterol biosynthesis. First, we checked the expression levels of *SREBP1* since the pathway “regulation of cholesterol biosynthesis by SREBP” was as significantly affected as cholesterol biosynthesis (similar q-values). This gene regulates transcription of key enzymes that participate in cholesterol biosynthesis namely HMG-CoA synthase (*HMGCS1*) and HMG-CoA reductase (*HMGCR*). Nevertheless, expression of *SREBP* was not conclusive as it was found upregulated in colon and not significantly affected in SI. Nevertheless, as shown in Appendix A, the encoding genes *HMGCS1* and *HMGCR* were downregulated in colon and upregulated in SI after the exposure. Next, we investigated if ER stress could have impacted cholesterol biosynthesis, as reported previously [33], but pathway and gene expression analysis showed that gefitinib did not induce ER stress in the organoids. 

We also evaluated the expression of *PRKAB1*, which regulates metabolism via AMPK, depending on p53 signalling activity. The *PRKAB1* gene was found upregulated in colon and downregulated in SI. This could indicate that AMPK became activated in colon unlike in SI, which could explain the differences in the metabolic responses upon TKI-induced stress, namely the contrast in the cholesterol *de novo* synthesis and metabolism of lipids. Further differences between colon and SI organoids were the expression levels of genes encoding CYP51A1 and ABCA1. The first is a crucial enzyme in cholesterol biosynthesis and expression levels were decreased in colon and increased in SI. In contrast, *ABCA1*, a membrane transporter responsible for cholesterol efflux in the form of HDL, was upregulated in colon organoids only. These metabolic genes are also indicated in Table 3.

Taken all together, gefitinib off-target metabolic processes were differently affected in both organoid types. Drug metabolism prevailed in SI as *CYP3A5* was upregulated across treatment conditions whereas, whereas *CYP3A4* was downregulated in colon. Energy production pathways were more significantly affected in colon, involving TCA cycle and ATP synthesis exclusively in these organoids. DEGs involved in these pathways were downregulated over time and concentrations except genes involved in the metabolism of lipids and cholesterol biosynthesis, as these were upregulated in SI in contrast to colon. These novel and promising observations emphasize the different responses of colon and SI organoids to gefitinib. Figure 6 also provides an overview of the effects of gefitinib in the expression of key genes involved in metabolism, namely in cholesterol biosynthesis, highlighting the differences between colon and SI organoids.

## 3. Discussion

Oral TKIs including gefitinib are becoming more common in the treatment of some cancers, such as NSCLC, renal cell carcinoma, metastatic breast cancer or colorectal cancer [37]. However, the side effects that this class of drugs may cause, particularly intestinal damage and diarrhoea [10], have been underestimated and are mechanistically not fully understood. 

In this study, we investigated the molecular mechanisms underlying the toxic effects of gefitinib in a newly established 3D culture of healthy human intestinal organoids derived from colon and SI tissues. These in vitro models have been developed to better resemble the characteristics and responses of the human tissues [18,38,39] and they have been successfully applied in the research of drug toxicity induced by 5-FU [14] and doxorubicin [40]. PBPK modelling was used in order to determine in-vitro drug concentrations that are clinically relevant. Overall, the cytotoxicity and transcriptomic data corroborates the gefitinib inhibitory effect on EGFR kinase signalling pathways and on intestinal cells growth. EGFR-regulated genes (*AURKA*, *CCND1*, *c-MYC*, *c-FOS* and *FGF19*) were downregulated, hampering cell cycle progression and cell differentiation, in line with the decreased cell viability and increased organoid roundness reflecting reduced differentiation of the organoids. Adhesion-related genes were downregulated as previously reported [33,34]. Furthermore, p53 and FOXO-regulated genes involved in cell cycle arrest and pro-apoptotic processes were upregulated, whereas anti-apoptotic genes were downregulated, particularly in colon organoids, which also presented a higher percentage of dead cells than SI. Novel and promising findings were metabolism related, showing more organ specific responses. Drug metabolism prevailed in SI with upregulation of CYP3A5, whereas in colon it was downregulated. Additionally, metabolism of lipids and cholesterol biosynthesis-related genes showed opposite direction of expression in colon and SI organoids after the exposure. These findings emphasize tissue-specific responses as colon seems to be more sensitive to the drug effects and SI seems to be more resistant. These gefitinib-induced gene expression changes are highlighted in Figure 6.

As mentioned previously, the known on-target mechanism of action for gefitinib is the selective blockade of EGFR-tyrosine kinase domain, preventing ATP from binding to it. As a result, EGFR autophosphorylation is inhibited as well as the activation of the signalling cascades associated with the receptor that would promote cell proliferation, growth and inhibition of apoptosis [3] (Figure 6). In our study, we observed the anti-proliferative ad apoptotic effects of gefitinib on the intestinal organoids, starting with the cytotoxicity assays and morphological changes. Gefitinib caused a decrease in cell viability and induction of apoptosis in both organoids, in a concentration- and time-dependent manner. Differences in the cytotoxicity results between the organoids were reflected in the viability being decreased already for the lowest dose, at 72 h, in SI organoids, but caspase 3/7 activation being increased more significantly and earlier in colon organoids (Figure 3). Likewise, image-based analysis (Figure 4) showed the drug effect on size and shape, which were more significantly perturbed in SI organoids, whereas percentage of cell death was more noticeable in colon organoids after exposure to 30 µM. The observations in the caspase activation assay were in line with the percentage of cell death. Nevertheless, pathway analysis showed that induction of apoptosis was not significantly affected throughout the treatment conditions. 

Pathway analysis on genes that were differentially expressed after exposure showed that on-target signalling transduction cascades and gene expression regulators p53 and FOXO were affected by gefitinib, being the first to be more significantly affected in colon, and p53 and FOXO signalling pathways more affected in SI organoids (Table 2). Downstream genes regulated by these signalling cascades and involved in cell cycle progression, cell growth and proliferation were downregulated, including *CCND1*, *c-MYC*, *AURKA*, *AREG*, *FGF19*, and tissue-specific *c-FOS* and *MYBL2* (Table 2). Cyclin D1 (*CCND1*) and p27 have been reported as downregulated in a previous study with gefitinib-treated IEC [33]. In our study, neither p27 nor p21 were found as modulated genes, which could indicate that healthy intestinal organoids are sensitive to gefitinib through a mechanism dependent on Cyclin D1, c-MYC and p53 rather than p27/p21. In fact, similar findings were reported in a study with colon cancer cell lines HCT116 treated with TKIs [41]. Furthermore, increased expression of *AREG* and *AURKA* in lung cancer cells has been associated with resistance to gefitinib treatment [42,43]. Remarkably, *FGF19* was found downregulated in cases of chronic diarrhoea caused by bile acid malabsorption [35], thus this gene could be an indicator of increased risk of diarrhoea in cancer patients taking gefitinib. Furthermore, DNA repair and damage/oxidative stress response genes *PCNA* and *SESN1*, in colon, and *DDB2*, in SI, were upregulated, which can represent a response of the organoids against the damaging effects of the drug. In contrast, downstream genes involved in cell cycle arrest, supressing proliferation, and activation of apoptosis were mostly upregulated. This relates to *BCL2L11*, *BTG2*, *MUC20*, and tissue-specific response genes *BAX*, *CCNG2* and *SUSD6* (Table 3). In previous studies, overexpression of *BCL2L11* and *BAX* have been associated with higher sensitivity of cancer cells to gefitinib [31,32]. 

Gefitinib also decreased the expression of adhesion molecules E-cadherin and β1-integrin, even though the effect on β1-integrin was only observed in colon organoids. Likewise, two previous studies on gefitinib and other TKIs corroborate these findings [33,34]. Levels of *PERP*, important in the maintenance of epithelial integrity and cell–cell adhesion, were decreased by gefitinib, which could aggravate the damaging effects of the drug. Moreover, the balance of the levels of *TNS3* and *TNS4* was disrupted by gefitinib effects on EGFR, hindering cell motility/migration. 

Gefitinib off-target immune responses were significantly affected in colon organoids (Table 3), particularly IL-6, IL-1 and IFN-ɣ signalling pathways at later treatment conditions. However, gene expression levels of *IL6*, *IL25* and *IFNG* were not significantly changed, whereas *IL1* was downregulated, which is in contrast with previous studies [33,34]. In addition, levels of toll-like receptor encoding gene *TLR2* were increased. Nevertheless, this is not an indication that inflammatory responses were increased, since no immune system cells were present during the exposure. Further studies are necessary with an in vitro model that includes immune cells (e.g., organoid-immune cell co-culture) to better understand how intestinal epithelial cells represented by organoids communicate with the local immune system, resulting in gefitinib-induced acute and/or chronic inflammation clinically. 

Moreover, exposure to gefitinib had an impact on metabolism in both types of organoids (Appendix A). Overall, energy production metabolic pathways including glycolysis, TCA cycle, pyruvate metabolism and respiratory electron chain/AT synthesis were more perturbed in colon organoids. DEGs involved in these pathways were mostly downregulated, reflecting the negative effect of gefitinib on the cells’ energy balance. In turn, effects on drug metabolism mediated by CYP450 were more prominent in SI organoids, particularly with the upregulation of *CYP3A5*. CYP450 enzymes play an important role in gefitinib metabolism and its clearance, lowering the drug’s concentration and, consequently, increasing resistance to the drug [4]. Noteworthily, studies have reported that protein levels of CYP3A are generally higher in SI than in colon [44,45]. Therefore, this could explain SI organoids lower sensitivity to gefitinib as compared with colon organoids. Metabolism of proteins, amino acids and lipids, as well as cholesterol biosynthesis were significantly affected in both types of organoids, but exposure concentration seemed to have more impact in colon whereas time of exposure had more impact in SI. An interesting finding was the different expression profiles of DEGs involved in the metabolism of lipids, important for energy production, and cholesterol biosynthesis, an essential component of cell membranes [46]. In colon, genes were clearly downregulated whereas in SI they were upregulated. After further investigation, we found that expression levels of *PRKAB1*, encoding for the regulatory subunit of AMPK and thus regulating its function, were upregulated in colon and downregulated in SI. AMPK is known to become activated and supress cell growth when cells are in stress and energy balance needs to be restored [47]. One of AMPK’s roles in response to cellular metabolic stresses is to phosphorylate and inactivate acetyl-CoA carboxylase (ACC) and β-hydroxy-β-methylglutaryl-CoA reductase (HMGCR), which are key enzymes of the biosynthesis of lipids and cholesterol, respectively [47]. Therefore, this is a strong indicator that AMPK activity was regulated differently in colon and SI due to *PRKAB1* expression, resulting in opposite metabolism of lipids and cholesterol. In addition, levels of *CYP51A1* and *ABCA1* were also opposite in the two different organoid models (Table 3), suggesting that cholesterol biosynthesis might be activated in SI in contrast to the colon where the efflux of cholesterol is enhanced.

Taken all together, these results show that colon organoids were more sensitive to exposure to gefitinib as compared with SI organoids, and the colon is therefore more likely to be damaged and lose epithelial integrity. This negative impact on the integrity of the colon’s epithelial barrier may explain the clinical side effects of gefitinib, namely diarrhoea. On the other hand, SI organoids seemed to possess a mechanism to better resist gefitinib’s effects by increasing cholesterol synthesis, unlike colon. In fact, some studies have reported that higher levels of cholesterol in the cellular membrane where EGFR is located, is associated with less sensitivity to gefitinib [48,49,50]. This novel finding brings promise to the improvement of cancer therapies in which increase of cholesterol levels in the colon tissue via co-administration may protect against the drug’s toxicity. 

Future studies including organoids derived from different donors and patients, as well as measurement of protein levels involved in key response mechanisms identified in this study, are needed to confirm these results and if they are indeed tissue-specific or donor-specific. Nevertheless, generating healthy colon and SI organoids from the same individual remains either an ethical challenge, as donors would need to undergo unnecessary surgical procedures, or a commercial one, as paired organoid models are not available yet. Once these challenges are overcome, the inclusion of paired healthy colon and SI organoids will be possible in future studies, so that donor genetic background variability, which influences drug-gene responses, is investigated. Moreover, transcriptomic data generated from intestinal tissue biopsies of cancer patients taking gefitinib as treatment is important to validate the molecular mechanisms of gefitinib-induced intestinal toxicity in the organoids. Well-validated markers of toxicity may eventually be applied in clinical settings to inform personalized therapies aiming to reduce the severity of drug-induced side effects. The new data generated from this study holds promise in supporting the development of tools to assess drug safety and de-risking strategies for novel therapeutic candidates before entering first-in-human trials. Furthermore, this data will be integrated in the development of computational GI drug toxicity prediction models being developed in the context of the transQST project [26].

## 4. Materials and Methods

### 4.1. Three-Dimensional In Vitro Culture of Human Healthy Intestinal Organoids

Human healthy colon and small intestine (SI) organoids were kindly provided by Boehringer Ingelheim Pharmaceuticals Inc. (Ridgefield, CT, USA) who purchased two tissue biopsies collected from healthy 67- and 74-year-old male donors, respectively, by Discovery Life Sciences (formerly, Conversant Biologics Inc., Huntsville, AL, USA) under the bio-specimen purchase agreement. All tissue samples were collected with written informed consent under an approval of the institutional review board (IRB). Noteworthily, the choice of healthy male donors to generate the organoids was based on tissue availability and did not consider the prevalence of adverse effects caused by gefitinib experienced by male and female patients. Cultivation and differentiation of the colon and SI tissue models was adapted from the methods described by Sato et al. [39]. Frozen organoids were recovered on a 24-well plate and grown in complete crypt medium [14] in order to maintain a stem cell state for propagation. Colon and SI cultures were passaged every 3–7 days as described in our previous work [14]. Prior to experimentation, organoids were transferred to 96-well plates and cultured in Human IntestiCult™ Organoid Growth Medium (Stemcell, Cologne, Germany) to promote cell differentiation. After 2–3 days, organoids started to differentiate and to exhibit intestine-like crypts and villi [14].

### 4.2. Selection of Gefitinib In Vitro Concentrations Based on PBPK Simulation

The Simcyp^®^ simulator (Version 19 release 1; Certara UK Ltd., Sheffield, UK) was used to develop and verify a model describing human gefitinib pharmacokinetics (PK) following oral dosing based on gefitinib physiochemical properties and relevant absorption, distribution, metabolism, and excretion (ADME) properties. The advanced dissolution, absorption, and metabolism (ADAM) model [51,52] was used to predict gefitinib concentrations in enterocytes in addition to concentrations in plasma. The details of parameters and sources used are listed in Appendix A. The data used to verify the model performance [27] were not used for model parameterisation. The verified gefitinib PBPK model was then used to predict plasma and jejunum enterocyte drug concentrations in healthy volunteers (aged 20–50 years old, 50% female, *n* = 100) after the therapeutic regimen of once daily 250 mg oral dose [28], over the course of 14 days. 

Simcyp’s in vitro data analysis toolkit (SIVA Version 3, module 3; Certara UK Ltd., Sheffield, UK) for virtual in vitro distribution (VIVD) [53] was used to simulate the distribution of gefitinib in human intestinal organoids in vitro based on gefitinib physiochemical properties, experimental design, and intestinal cell composition from Simcyp V19r1. In vitro nominal gefitinib concentrations that achieved intracellular concentrations equivalent to those predicted using PBPK was explored. The input parameters for this model are detailed in Appendix A.

### 4.3. Design of In Vitro Exposure to Gefitinib

Gefitinib was purchased from Merck (Darmstadt, Germany), with ≥99% purity. Selection of gefitinib concentrations was based on the PBPK calculation methods described above. After 3 days in Human IntestiCult Growth medium, differentiated intestinal organoids were exposed to 100 µL of medium containing 0.1 µM, 1 µM, 10 µM and 30 µM gefitinib for 24 h, 48 h and 72 h. Controls containing only IntestiCult medium (untreated controls) and IntestiCult medium with 0.1% DMSO (vehicle controls) were included for all time points. All treatment conditions were performed in biological triplicates in 96-well plates. The remaining wells were filled with 100 µL PBS to eliminate possible edge effects. For all treatment conditions, samples were collected to evaluate toxicity assays and perform RNA sequencing.

### 4.4. Cytotoxicity Assays: ATP-Based Viability and Caspase 3/7 Activity Evaluation 

Cytotoxicity endpoints for viability and caspase 3/7 activity were measured with 3D Celltiter-Glo and Caspase-Glo 3/7 (Promega, Madison, WI, USA), respectively, per manufacturer’s instructions. These endpoints were used to evaluate the toxicity profile of gefitinib and to check if they were reflected on the transcriptomic data. After each time point of exposure, medium was removed and replaced by 100 µL of either kit reagent to the appropriate wells and well homogenized. The plates were placed in a Scilogex MX-M 96 well plate shaker for 1 h at room temperature. Afterwards, samples were transferred to white opaque 96-well plates (Corning, NY, USA) to measure luminescence values in GloMax^®^ 96 Microplate Luminometer (Promega, Madison, WI, USA). Luminescence values were transferred to GraphPad Prism 9.0 (GraphPad Software, San Diego, CA, USA), normalized to the respective time-controls, and corrected for the blank reaction to eliminate possible interferences of the matrigel in the absorbance. Statistical differences between conditions were calculated by applying the analysis of variance (ANOVA) test.

### 4.5. Image Analysis of Morphological Changes

Colon and SI organoids were grown and treated with the same conditions as described previously, after which changes in morphology and structure of the cells were analysed. This analysis aimed to support the cytotoxicity evaluation and to provide insight into significant structural changes associated with gefitinib’s mechanism of toxicity. After each time point, fixation in 3% Formaldehyde and staining were performed to visualize the nuclei (in blue) and actin cytoskeleton (in red), by applying a solution containing Hoechst 33258 (Merck, Darmstadt, Germany) with a final concentration of 0.4 µg/mL, and Rhodamine-phalloidin (Merck, Darmstadt, Germany), with final concentration of 0.1 µM [54]. Images were captured as z-stacks in an ImageXpress Micro XLS (Molecular Devices, Silicon Valley, CA, USA) wide field microscope, using the 4× objective, and analysed in the 3D image analysis solution Ominer^®^ (Crown Bioscience Netherlands B.V., Leiden, the Netherlands). GraphPad Prism 9.0 (GraphPad Software, San Diego, CA, USA) was used to obtain values for average organoids size, roundness and dead cells. The treatment values were normalized to the respective time-controls.

### 4.6. RNA Isolation and Quantification

For each treatment condition, medium was removed and 200 µL of QIazol Lysis reagent (Qiagen, Venlo, The Netherlands) was added into each well, in which the matrigel was dissociated to collect each pellet. This process was repeated twice, ensuring collection of all organoids as well as ensuring a total volume of 700 µL of QIazol Lysis reagent in each tube containing pellet. An extra vigorous pipetting and vortex was performed to allow complete homogenization of organoids in the lysis reagent. RNA isolation and purification were performed using the miRNeasy Mini Kit (Qiagen, Venlo, The Netherlands), following manufacturer’s protocol for Animal Cells including a DNase treatment. Total RNA yield was measured on Nanodrop^®^ ND-1000 spectrophotometer (Thermo Fisher Scientific, Waltham, MA, USA). RNA quality was checked with RNA Nanochips or Picochips on a 2100 Bioanalyzer (Agilent Technologies, Leuven, Belgium). All samples presented integrity number (RIN) > 7.5 and average RNA yield of 607 ng, for colon samples, and 262 ng, for SI samples.

### 4.7. Library Preparation and mRNA Sequencing

Purified RNA from each sample was prepared for sequencing using the Lexogen SENSE mRNA library preparation kit (Lexogen, Vienna, Austria), following the manufacturer’s instructions. Quality checks were assessed for every library. Afterwards, the samples were sequenced on the NovaSeq 6000 system (Illumina, Eindhoven, The Netherlands). A pool of all treated and untreated samples was sequenced on the two lanes of a S1 flow cell. The average gene count was 14.58 million raw reads. 

### 4.8. Data Processing and Analysis 

The first step of pre-processing consisted in the removal of the 12 bases of the 5′end of all reads, which correspond to the Lexogen adapter sequences, using Trimmomatic version 0.33 [55]. Before and after this trimming step, the quality of the sequencing data was confirmed using FastQC version 0.11.3 [56] and only samples that met the required parameters were used for subsequent analysis. Following trimming and QC check, gene reads were aligned to the complete human genome (Ensembl build v. 93 GRCh38) using Bowtie 1.1.1 and quantified with RSEM 1.3.1, with an average of 10 million reads per sample. Normalization of the quantified read counts from all samples was performed using the R package DESeq 2 (v. 1.14.1) (Bioconductor, Seattle, WA, USA) [57] resulting in a list of around 13,000 differentially expressed genes (DEGs) for each treatment condition. Moreover, the profile and distribution of the samples were obtained according to the amount of reads, hierarchical clustering, principal component analysis (PCA) and sample dispersion. For each time point, the following comparisons were possible: (a) Untreated control vs. Vehicle control; (b) 0.1 µM vs. Vehicle control; (c) 1 µM vs. Vehicle control; (d) 10 µM vs. Vehicle control; (e) 30 µM vs. Vehicle control. In addition, Bonferroni correction [30] was applied to all genes obtained, after which genes with adjusted *p*-value < 0.05 were considered as differential expressed genes (DEGs). 

### 4.9. Pathway Analysis Based on DEGs

The lists of DEGs obtained for each time point and concentration were used as input for pathway overrepresentation analysis (ORA) using ConsensusPathDB (CPDB) release 34 [58], considering a cut-off of 0.01. ORA analysis provided an overview of biological pathways affected in treated samples as compared to vehicle controls. The Reactome database version 67 [59], WikiPathways [60] and KEGG [61] were selected as preferred databases for pathway analysis and interpretation of biological processes. The most significantly overrepresented pathways were identified using the q-values (q < 0.05) and the number of DEGs involved. The most relevant DEGs we visually summarised using BioRender illustration tool [62].

## Figures and Tables

**Figure 1 ijms-23-02213-f001:**
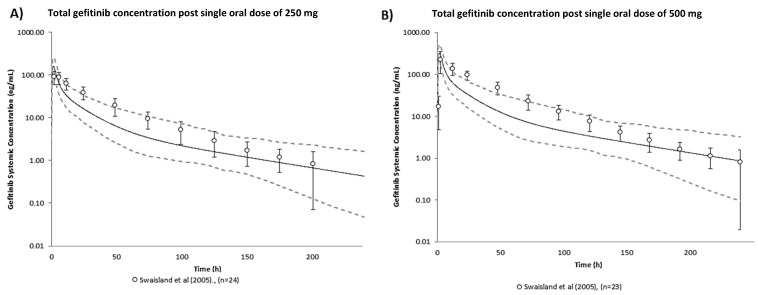
Verification of the gefitinib human PBPK model: predicted geometric mean (solid black line) and observed geometric mean (data points) ± standard deviation (error bars) of total gefitinib concentration post single oral dose of (**A**) 250 mg and (**B**) 500 mg [27]. The dashed lines refer to the predicted 95th percentile range. Virtual population demographic was simulated to be equivalent to the reported study (healthy volunteers, aged 21–57 years old, 0% female, *n* = 100).

**Figure 2 ijms-23-02213-f002:**
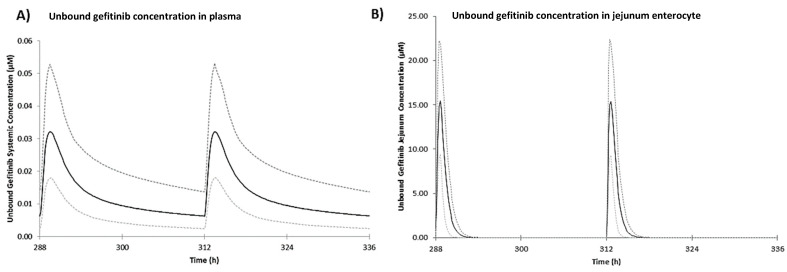
Predicted mean (solid black line) of unbound gefitinib concentration in (**A**) plasma and (**B**) jejunum enterocyte for the last two doses during once daily 250 mg oral dose in human over 14 days. The dashed lines refer to the 95th percentile range of the simulated virtual population (healthy volunteers, aged 20–50 years old, 50% female, *n* = 100).

**Figure 3 ijms-23-02213-f003:**
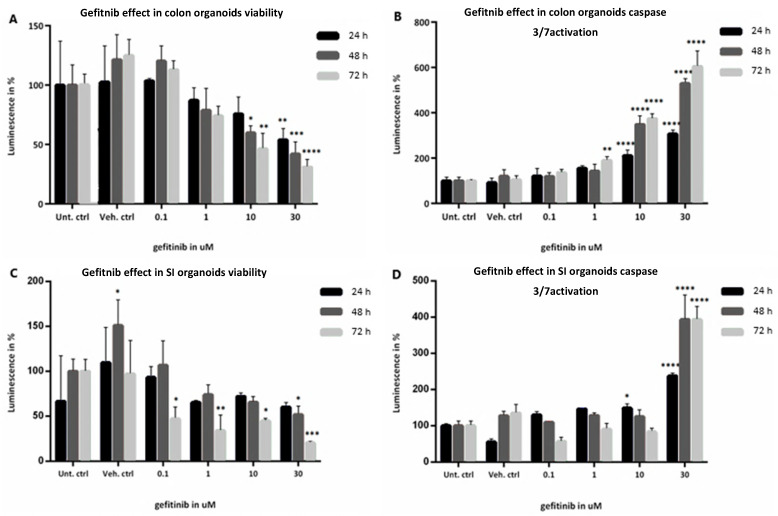
(**A**–**D**): Evaluation of viability determined by ATP levels and caspase 3/7 activation of healthy colon (**A**,**B**, respectively) and SI (**C**,**D**, respectively) organoids when exposed to 0.1, 1, 10, and 30 µM gefitinib for 24 h in black, 48 h in dark grey and 72 h in light grey, compared with Untreated controls. Values are in % of luminescence. For each time point the average of Unt. Ctrl was set to 100%. SD was calculated for each condition. Legend: Ctrl, control; SD, standard deviation; SI, small intestine; Unt, untreated; Veh, vehicle (with 0.1% DMSO). * *p*-value = 0.01; ** *p*-value = 0.003; *** *p*-value = 0.0004; **** *p*-value = 0.0001.

**Figure 4 ijms-23-02213-f004:**
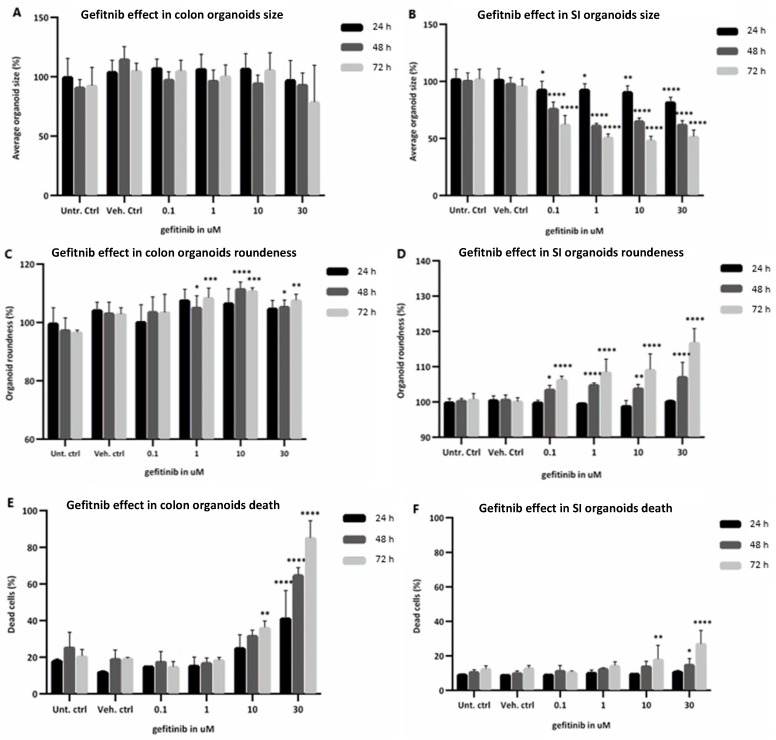
Morphological changes assessed through imaging analysis of healthy colon (**A**,**C**,**E**) and SI (**B**,**D**,**F**) organoids when exposed to 0.1, 1, 10 and 30 µM gefitinib for 24 h in black, 48 h in light grey and 72 h in dark grey, compared with untreated controls. (**A**,**B**): organoid size; (**C**,**D**): organoid roundness; (**E**,**F**): percentage of dead cells. Values are in % based on fluorescent intensity for each measured parameter. SD was calculated for each condition. Legend: Ctrl, control; SD, standard deviation; SI, small intestine; Unt, untreated; Veh, vehicle. * *p*-value = 0.01; ** *p*-value = 0.008; *** *p*-value = 0.0008; **** *p*-value = 0.0001.

**Figure 5 ijms-23-02213-f005:**
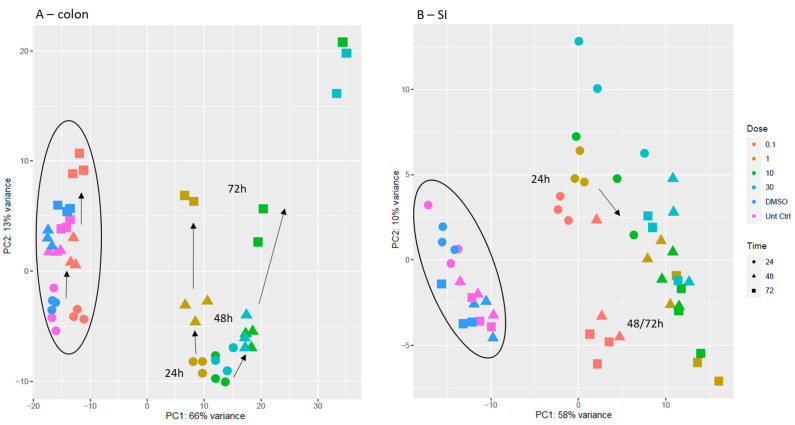
PCA score scatter plot obtained for the mRNA transcriptomic analysis of samples collected from colon organoids (**A**) and SI organoids (**B**). Direction of the arrows indicates the evolution in time of the samples (24 h → 48 h → 72 h). In colon plot, cluster on the left comprises non-treated samples (untreated and vehicle controls); in SI plot, cluster on the left comprises non-treated samples and treated samples with 0.1 µM gefitinib. Legend: untreated controls are in pink; vehicle controls (DMSO) are in dark blue; 0.1 µM gefitinib in red; 1 µM gefitinib in dark yellow; 10 µM gefitinib in green; and 30 µM gefitinib in light blue. Circles represent 24 h; triangles, 48 h; squares, 72 h.

**Figure 6 ijms-23-02213-f006:**
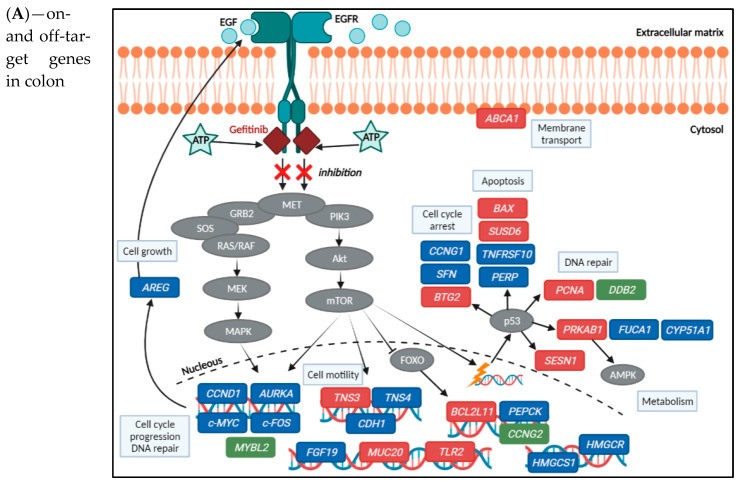
Representation of gefitinib effects on expression levels of on- and off-target genes in the colon (**A**) and SI (**B**) organoids, elucidating the drug’s potential mechanisms of toxicity. For the alterations in the gene expression levels, all treatment conditions were considered. Legend: genes in blue—significantly downregulated; red—significantly upregulated, after Bonferroni correction (adjusted *p*-value < 0.05); green—not available/not significant; grey—protein expression not available. Image created with BioRender.com (accessed on 26 June 2021).

**Table 1 ijms-23-02213-t001:** **VIVD-Based Predictions of in Vitro Human Intestinal Organoid Intracellular Concentrations at Steady State for Different in Vitro Nominal Concentrations.** PBPK-predicted in vivo mean gefitinib maximum concentration at steady state (C_max,ss_), minimum concentration at steady state (C_min,ss_) and average concentration at steady state (C_av,ss_) for once daily dosing of 250 mg oral gefitinib in healthy volunteers (aged 20–50 years old, 50% female, *n* = 100).

**VIVD-based in vitro predictions**			
Nominal concentration (µM)	Human intestinal organoid intracellular concentration (µM)		
**0.01**	0.91		
0.1	9.14		
1	91.37		
10	913.67 ^†^		
**PBPK-based in vivo predictions during 250 mg once daily dosing**			
	C_av,ss_ (µM)	C_max,ss_ (µM)	C_min,ss_ (µM)
Total plasma	0.31	0.51	0.10
Unbound plasma	0.02	0.03	0.01
Total and unbound enterocyte *	7.69	15.38	5.98 × 10^−14^

^†^ Prediction based on assumption of non-saturable binding, hence the ratio of nominal to intracellular concentration remains constant; * total and unbound gefitinib in enterocytes assumed equal.

**Table 2 ijms-23-02213-t002:** Most relevant on-target pathways perturbed by gefitinib.

Name of the Pathway	Pathway Source	Time (h)	Gefitinib Concentration (µM)	q-Value/Number of DEGs	
				Colon	SI
EGF/EGFR signalling pathway	Reactome	24	0.1	*NA*	*NA*
			1	*NA*	*NA*
			10	*NA*	*NA*
			30	*NA*	*NA*
		48	0.1	*NA*	*NA*
			1	*NA*	*NA*
			10	*NA*	*NA*
			30	*NA*	*NA*
		72	0.1	*NA*	*NA*
			1	*NA*	*NA*
			10	**0.01/13**	*NA*
			30	**0.01/17**	*NA*
PI3K-Akt signalling pathway	KEGG	24	0.1	**0.04/2**	*NA*
			1	**0.001/38**	**0.04/11**
			10	**1.0 × 10^−4^/48**	*NA*
			30	**5.0 × 10^−4^/51**	**0.01/16**
		48	0.1	*NA*	*NA*
			1	**0.002/14**	0.13/14
			10	**3.44 × 10^−5^/55**	0.1/13
			30	**4.98 × 10^−5^/54**	*NA*
		72	0.1	*NA*	*NA*
			1	0.09/18	**0.05/23**
			10	**0.04/58**	*NA*
			30	0.08/82	**0.04/17**
MAPK signalling pathway	Reactome	24	0.1	*NA*	*NA*
			1	**0.01/25**	**0.02/10**
			10	**0.003/31**	*NA*
			30	**0.01/32**	*NA*
		48	0.1	*NA*	*NA*
			1	0.14/20	0.15/10
			10	**0.03/31**	0.15/9
			30	**0.04/29**	*NA*
		72	0.1	*NA*	*NA*
			1	*NA*	0.08/16
			10	**0.02/44**	*NA*
			30	**0.04/60**	*NA*
Signalling by MET	Reactome	24	0.1	*NA*	*NA*
			1	**0.002/12**	0.14/3
			10	**0.003/13**	*NA*
			30	**0.02/12**	*NA*
		48	0.1	*NA*	*NA*
			1	**0.002/12**	*NA*
			10	**0.004/14**	*NA*
			30	**0.03/14**	*NA*
		72	0.1	*NA*	*NA*
			1	**0.02/7**	*NA*
			10	**4.0 × 10^−4^/20**	*NA*
			30	**1.22 × 10^−5^/29**	*NA*
Signalling by NOTCH	Reactome	24	0.1	*NA*	0.06/4
			1	**0.01/16**	**0.005/8**
			10	**0.003/20**	0.1/5
			30	**0.005/21**	*NA*
		48	0.1	*NA*	*NA*
			1	*NA*	0.06/8
			10	0.16/15	*NA*
			30	0.09/16	*NA*
		72	0.1	*NA*	*NA*
			1	0.06/9	*NA*
			10	**0.005/28**	*NA*
			30	**1.28 × 10^−5^/46**	*NA*
FOXO-mediated transcription	Reactome	24	0.1	*NA*	*NA*
			1	**0.04/15**	*NA*
			10	**0.01/19**	*NA*
			30	0.06/18	**0.0003/12**
		48	0.1	*NA*	*NA*
			1	*NA*	0.07/8
			10	0.12/17	**0.003/10**
			30	*NA*	**0.02/10**
		72	0.1	*NA*	*NA*
			1	*NA*	**0.007/14**
			10	*NA*	**0.002/12**
			30	**0.05/36**	**0.04/9**
p53 signalling pathway	KEGG	24	0.1	*NA*	*NA*
			1	**0.009/12**	*NA*
			10	**0.02/12**	*NA*
			30	0.06/12	**0.0003/9**
		48	0.1	*NA*	*NA*
			1	0.09/9	0.06/6
			10	**0.02/14**	*NA*
			30	0.06/12	**0.0005/10**
		72	0.1	*NA*	*NA*
			1	*NA*	**0.02/9**
			10	*NA*	**0.001/9**
			30	*NA*	**0.001/9**
Regulation of mitotic cell cycle	Reactome	24	0.1	*NA*	*NA*
			1	**0.01/8**	*NA*
			10	**0.001/11**	*NA*
			30	**0.003/11**	**0.04/4**
		48	0.1	*NA*	*NA*
			1	*NA*	*NA*
			10	*NA*	*NA*
			30	*NA*	*NA*
		72	0.1	**0.01/2**	*NA*
			1	*NA*	*NA*
			10	0.13/9	*NA*
			30	0.09/13	*NA*
Cellular senescence	KEGG	24	0.1	*NA*	*NA*
			1	**0.01/19**	*NA*
			10	0.06/16	*NA*
			30	**0.004/26**	**0.01/10**
		48	0.1	*NA*	*NA*
			1	0.012/15	*NA*
			10	**0.05/22**	*NA*
			30	**0.01/24**	**0.003/13**
		72	0.1	*NA*	*NA*
			1	*NA*	**0.002/17**
			10	*NA*	**3.0 × 10^−4^/15**
			30	0.14/39	**3.0 × 10^−4^/15**
Apoptosis	Reactome	24	0.1	*NA*	*NA*
			1	*NA*	*NA*
			10	0.11/14	*NA*
			30	*NA*	*NA*
		48	0.1	*NA*	*NA*
			1	*NA*	0.09/7
			10	0.15/15	*NA*
			30	0.13/15	*NA*
		72	0.1	*NA*	*NA*
			1	*NA*	**0.008/13**
			10	0.06/23	*NA*
			30	**0.05/33**	**0.02/9**

Legend: significant q-values < 0.05 (in bold) or not applicable (NA) when the respective pathways were not present; KEGG, Kyoto Encyclopaedia of Genes and Genomes.

**Table 3 ijms-23-02213-t003:** Expression changes of the DEGs affected by gefitinib in colon and SI organoids.

Gene Symbol	Pathways Involved/Function	Expression in Treated Colon/SI Cells (Adjusted *p*-Value) *	Remarks
**DEGs with same trend in colon and SI organoids**			
** *AREG* **	Interaction with EGFR to promote growth of epithelial cells	↓ (9.7 × 10^−8^)/↓ (6.5 × 10^−7^)	
** *AURKA* **	DNA repair; Cell cycle	↓ (3.9 × 10^−6^)/↓ (0.04)	
** *BAX* **	Induction of apoptosis	↑ (1.5 × 10^−4^)/↑ (1.0)	Activated in gefitinib treated gallbladder cancer cells [31]
** *BCL2L11 (BIM)* **	Induction of apoptosis	↑ (2.1 × 10^−5^)/↑ (1.1 × 10^−8^)	↑ linked to higher sensitivity to gefitinib in NSCLC [32]
** *BTG2* **	Cell cycle arrest (G1/S)	↑ (5.5 × 10^−29^)/↑ (7.1 × 10^−6^)	
** *c-MYC* **	Cell cycle progression	↓ (8.1 × 10^−30^)/↓ (1.7 × 10^−10^)	
** *CCND1* **	Cell cycle/proliferation	↓ (0.03)/↓ (2.8 × 10^−32^)	↓ in gefitinib treated IEC [33]
** *CCNG1* **	Cell cycle arrest	↓ (5.5 × 10^−6^)/↓ (1.0)	
** *CCNG2* **	Cell cycle arrest	↑ (1.0)/↑ (0.004)	
** *CDH1* **	E-cadherin; cell adhesion molecule	↓ (8.5 × 10^−8^)/↓ (1.5 × 10^−5^)	↓ in TKI-treated IEC [33,34]
** *DDB2* **	DNA repair	↑ (0.13)/↑ (1.4 × 10^−4^)	
** *FGF19* **	Cell growth, tissue repair, effects on glucose and lipid metabolism	↓ (9.6 × 10^−9^)/↓ (0.02)	↓ in chronic diarrhoea [35]
** *ITGB1* **	β1-integrin; cell adhesion, embryogenesis, homeostasis, tissue repair, immune response	↓ (8.3 × 10^−17^)/↓ (1.0)	↓ in TKI-treated IEC [34]
** *MUC20* **	Suppressor of MET signalling/proliferation	↑ (9.0 × 10^−9^)/↑ (0.005)	
** *MYBL2* **	Cell cycle/proliferation	↓ (1.0)/↓ (6.1 × 10^−4^)	
** *PCNA* **	DNA repair	↑ (1.4 × 10^−8^)/↑ (1.0)	
** *PERP* **	Apoptosis effector; epithelial integrity and cell-cell adhesion	↓ (1.7 × 10^−16^)/↓ (0.03)	
** *SFN* **	Cell cycle arrest (G2/M)	↓ (1.1 × 10^−7^)/↓ (0.04)	
** *SUSD6 (KIAA0247)* **	Suppressor of cell growth; activator of apoptosis	↑ (8.8 × 10^−6^)/↑ (1.0)	
** *TLR2* **	Inflammatory signals	↑ (0.005)/↑ (1.0)	
** *TNFRSF10 (TRAIL-R2)* **	Induction of apoptosis	↓ (0.007)/↓ (1.0)	
** *TNS3* **	Cell motility/migration	↑ (6.5 × 10^−14^)/↑ (1.0)	
** *TNS4* **	Cell motility/migration	↓ (0.01)/↓ (5.8 × 10^−8^)	
**DEGs with opposite trend in colon and SI organoids**			
** *ABCA1* **	Efflux of cholesterol	↑ (0.02)/↓ (1.0)	
** *c-FOS* **	Cell cycle progression	↓ (3.2 × 10^−6^)/↑ (1.0)	
** *CYP51A1* **	Cholesterol biosynthesis	↓ (1.4 × 10^−22^)/↑ (2.2 × 10^−8^)	
** *DTX3* **	Regulator of Notch signalling	↑ (0.03)/↓ (1.0)	
** *FUCA1* **	Degradation of glycoproteins and glycolipids	↓ (8.7 × 10^−7^)/↑ (1.0)	
** *HMGCR* **	Cholesterol biosynthesis	↓ (4.5 × 10^−11^)/↑ (7.4 × 10^−4^)	
** *HMGCS1* **	Cholesterol biosynthesis	↓ (1.6 × 10^−4^ )/↑ (0.004)	
** *PEPCK* **	Glucose metabolism	↓ (3.1 × 10^−6^)/↑ (1.0)	
** *PRKAB1* **	AMPK-mediated metabolism	↑ (7.9 × 10^−10^)/↓ (0.008)	
** *SESN1* **	DNA damage and oxidative stress response; translation control	↑ (1.4 × 10^−7^)/↓ (1.0)	

Legend: * adjusted *p*-values (in brackets) above 0.05 were considered as not significant; ↑—upregulated; ↓—downregulated; IEC—intestinal epithelial cells; NSCLC—non-small cell lung cancer.

## Data Availability

The cytotoxicity data generated and analysed during the current study are available in the BioStudies repository (www.ebi.ac.uk/biostudies/studies/S-TQST115). The transcriptomic data generated and analysed during the current study will be publicly available on ArrayExpress repository (www.ebi.ac.uk/arrayexpress/) with accession number E-MTAB-11221.

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
