# Peer review of "A Transcriptomic Approach to Elucidate the Mechanisms of Gefitinib-Induced Toxicity in Healthy Human Intestinal Organoids"

_ijms, 2022, doi:10.3390/ijms23042213_

Round 1
Reviewer 1 Report
This work aims to understand the mechanisms involved in the induction of toxicity driven by gefitinib via a transcriptomic approach. It is an interesting manuscript. However, some questions need to be addressed.
The authors mentioned that the adverse effect caused by gefitinib can be around 30% of the patients. In addition, jumping to the material and methods, it was informed in the topic 4.1 “D in vitro culture…” (Is it supposed to be “3D in vitro culture…”?) that it was used biopsies from healthy old male donors. Why did the authors choose male for this study? Is there any prevalence between male and female around those 30% of the patients that present adverse effects caused by gefitinib? It needs to be explained.
The methods need to be improved, describing with more details how the fixation was performed, for example, and the text needs to be revised based on some misspelling. The way the authors organized the charts, and the legends is counterintuitive. Maybe adding title in each chart would facilitate the understanding and it would avoid having to check constantly the legend. Another thing that is concerning is the topic that addresses the morphological changes in the organoids by image-based analysis with no images of the organoids. Did the authors take pictures of the morphological changes in the organoids?
Reviewer 2 Report
This paper is interesting. The mechanism investigation by using organoids has been recently noted. However, the authors should introduce the 3D cell culture, such as spheroids or organoids, to evaluate cell function or mechanisms responding to each tissue. The readers must be confused. In addition, the authors should indicate the pictures of 3D organoids and show immunohistological findings with or without treatment. The readers cannot understand the situation. Taken together, major revisions should be made before re-submission. The paper would be re-considered only when all the comments were responded.
- Introduction
For the investigation of anticancer drug effects or performing the anticancer drug screening, 3D cell culture has been recently noted. The background is entirely lacking. The authors should add some sentences for the description of the fields. To reduce the authors’ burden, I suggest at least these recent papers be added for revision.
Brain (Review and research)
Annu. Rev. Neurosci. 2020. 43:375–89
Scientific Reports volume 9, Article number: 1407 (2019)
Cancer (Review and research)
Cancers 2020, 12(10), 2754
Tissue Eng. Part C Methods 2019, 25, 711–720 https://doi.org/10.1089/ten.tec.2019.0189
Lung (Review and research)
Nature Communications volume 10, Article number: 3991 (2019)
Cell Reports 27, 3709–3723, 2019.
- Results
The authors should indicate the pictures of 3D organoids and show immunohistological findings with or without treatment.
- Overall
The figures are a little difficult to see and understand.
- Figure 3
The luminescence is not good to show the ATP levels. I think cell numbers affect the results. Therefore, the ATP levels should be divided by cell number.
Round 2
Reviewer 2 Report
I recommend the publication.